# Capsaicin and Zinc Promote Glucose Uptake in C2C12 Skeletal Muscle Cells through a Common Calcium Signalling Pathway

**DOI:** 10.3390/ijms23042207

**Published:** 2022-02-17

**Authors:** Parisa Vahidi Ferdowsi, Kiran D. K. Ahuja, Jeffrey M. Beckett, Stephen Myers

**Affiliations:** School of Health Sciences, College of Health and Medicine, University of Tasmania, Newnham Drive, Launceston, TAS 7248, Australia; parisa.vahidiferdowsi@utas.edu.au (P.V.F.); kiran.ahuja@utas.edu.au (K.D.K.A.); jeffrey.beckett@utas.edu.au (J.M.B.)

**Keywords:** glucose metabolism, CAMKK2, CREB, TORC1, *Nr4a3*, *Junb*, calcium flux, cAMP signalling

## Abstract

Capsaicin and zinc have recently been highlighted as potential treatments for glucose metabolism disorders; however, the effect of these two natural compounds on signalling pathways involved in glucose metabolism is still uncertain. In this study, we assessed the capsaicin- or zinc- induced activation of signalling molecules including calcium/calmodulin-dependent protein kinase 2 (CAMKK2), cAMP-response element-binding protein (CREB), and target of rapamycin kinase complex 1 (TORC1). Moreover, the expression status of genes associated with the control of glucose metabolism was measured in treated cells. The activation of cell signalling proteins was then evaluated in capsaicin- or zinc treated cells in the presence or absence of cell-permeant calcium chelator (BAPTA-AM) and the CAMKK inhibitor (STO-609). Finally, capsaicin- and zinc-induced glucose uptake was measured in the cells pre-treated with or without BAPTA-AM. Our results indicate that calcium flux induced by capsaicin or zinc led to activation of calcium signalling molecules and promoting glucose uptake in skeletal muscle cells. Pharmacological inhibition of CAMKK diminished activation of signalling molecules. Moreover, we observed an increase in intracellular cAMP levels in the cells after treatment with capsaicin and zinc. Our data show that capsaicin and zinc mediate glucose uptake in C2C12 skeletal muscle cells through the activation of calcium signalling.

## 1. Introduction

Emerging studies on the importance of food components on glucose metabolism have highlighted capsaicin and zinc as potential therapeutic targets for carbohydrate metabolism diseases [1,2,3,4]. Capsaicin, the bioactive phenolic component of chilli peppers, has potential benefits in the reduction of glucose metabolism disorders and acts through the activation of transient receptor potential cation channel subfamily V member (TRPV1) [1,3]. Similarly, zinc plays an essential role in the prevention of carbohydrate metabolism diseases, and disruption in zinc homeostasis is strongly associated with the pathogenesis of these disorders [2,5].

Normal glucose homeostasis is critical for long-term health and is disrupted in metabolic diseases such as insulin resistance (IR) and type 2 diabetes mellitus (T2DM) [6]. Studies from animal models and human research support the effectiveness of capsaicin and zinc on the improvement of glucose metabolism [1,7,8,9]. Regular dietary capsaicin intake in mice increases plasma insulin levels and sensitivity, reduces inflammatory factors and blood glucose levels, which subsequently improves glucose homeostasis [1,10,11]. In humans, a chilli-containing diet causes a reduction in the amount of insulin required to control postprandial glucose levels [7]. Moreover, four weeks’ regular use of capsaicin-containing chilli supplementations enhances postprandial glucose metabolism in women with gestational diabetes mellitus [12]. It has also been shown that capsicum capsules (containing capsaicin) decrease plasma glucose concentrations and enhance plasma insulin levels in healthy individuals [13]. In the same way, zinc plays a significant role in insulin action and carbohydrate metabolism [14]. Zinc supplementation diminishes insulin resistance and blood glucose levels in obese high-fat diet mice [15]. Conversely, plasma zinc concentration is lower in diabetic patients compared to healthy people. This suggests that zinc deficiency is associated with the development of IR and T2DM [9,16].

Although extensive research demonstrates the beneficial effects of capsaicin and zinc on glucose homeostasis [1,7,9,17,18], there is limited information to describe the mechanism of action of these two food components in skeletal muscle glucose metabolism [2,3,19]. Capsaicin influences glucose metabolism independent of insulin signal transduction in mouse skeletal muscle cells [3,19]. Like capsaicin, the advantageous effects of zinc on glucose homeostasis are well-established in the literature [20,21]. Zinc stimulates glucose uptake through the activation of insulin signalling molecules in mouse skeletal muscle cells [2,22]. Based on our previous work, the central signalling pathways activated by capsaicin and zinc are distinct [2,3,22]; however, it appears that calcium signalling cascade is the common pathway between capsaicin and zinc in skeletal muscle cells [23,24]. Understanding the joint signalling pathway activated by capsaicin and zinc could potentially lead to the development of a combination therapy that is more efficient for the treatment of carbohydrate metabolism disorders.

Calcium is a ubiquitous second messenger regulating signalling pathways involved in glucose homeostasis in skeletal muscle [25,26,27]. Increased cytosolic calcium levels activate intracellular signalling molecules including CAMKK2 [28] and its downstream proteins including CREB and TORCs [29,30,31,32]. Activation of CREB and TORC1 promotes glucose uptake in skeletal muscle cells by regulating the expression of genes associated with glucose metabolism which include activating transcription factor 3 (*Atf3)*, V-jun avian sarcoma virus 17 oncogene homolog B (*Junb)*, and nuclear receptor 4 A3 (*Nr4a3)* [33,34]. Similar to intracellular calcium elevation, CREB, TORC1, and the downstream target genes can be activated in response to increases in cAMP levels in the cells [33,35].

While several studies partly described the molecular signals activated by capsaicin and zinc in glucose metabolism, the effect of calcium signal transduction on glucose uptake by these two bioactive food components is not fully investigated in skeletal muscle cells [1,2,3,19,22].

Accordingly, we aimed to investigate the effect of capsaicin and zinc on the calcium signalling pathway in C2C12 skeletal muscle cells. To the best of our knowledge, the present study is the first research to show that calcium signal transduction induced by capsaicin and zinc stimulates glucose uptake in C2C12 skeletal muscle cells. We have demonstrated that capsaicin and zinc activate CAMKK2, CREB, and TORC1 and regulate the expression of downstream genes. We observed an increase in intracellular cAMP concentration in the cells treated with capsaicin and zinc. Moreover, we found that reduction in cytosolic calcium levels significantly reduces activation of CAMKK2, CREB, and TORC 1 and subsequently decreases glucose uptake caused by capsaicin and zinc. Finally, we discovered that the pharmacological inhibition of CAMKK2 by STO-609 diminishes activation of CAMKK2, CREB, and TORC1 in C2C12 skeletal muscle cells.

## 2. Results

### 2.1. Capsaicin and Zinc Elevate Glucose Uptake in C2C12 Skeletal Muscle Cells

Normal blood glucose regulation is essential in maintaining appropriate cellular processes. IR and T2DM cause a reduction in glucose uptake levels resulting in hyperglycemia and the associated negative side effects in the body [1]. To delineate if capsaicin and zinc promote glucose uptake in skeletal muscle cells, we treated cells with 100 µM capsaicin, 20 µM ZnSO_4_ + 10 µM pyrithione sodium (NaPy) (we used NaPy to facilitate entering of zinc into the cells) or 10 nM insulin for 60 min. As displayed in Figure 1, 100 µM capsaicin, as well as 20 µM ZnSO_4_ + 10 µM NaPy, stimulates glucose uptake compared with the control group in skeletal muscle cells. Additionally, there is a significant elevation in glucose uptake in the cells treated with 10 nM insulin in comparison with the control group, which confirms the robustness of our system. 

### 2.2. Capsaicin and Zinc Stimulate Calcium Flux in C2C12 Skeletal Muscle Cells

Calcium plays an important role in glucose metabolism by stimulating glucose transporter type 4 (GLUT4) translocation to the cell membrane and the subsequent increase in glucose uptake in skeletal muscle cells [36]. To verify the effect of capsaicin or ZnSO_4_ on calcium flux and elevation of cytosolic calcium levels in C2C12 skeletal muscle cells, a calcium assay using fluorescent calcium indicator, Fluo-4, was utilised. First, to test the effect of capsaicin or zinc treatments on calcium flux from intracellular stores, we treated cells in a calcium-free Hanks’ balanced salt solution (HBSS) (Figure 2a,b) or HBSS buffer containing 2 mM CaCl_2_ (Figure 2c,d) with capsaicin or 20 µM ZnSO_4_ + 10 µM NaPy and compared calcium flux-induced by these two food components with the control group (untreated cells in time 0) over 300 s of incubation time. We found that 100 µM capsaicin or 20 µM ZnSO_4_ + 10 µM NaPy causes an elevation in cytosolic calcium levels after 30 s of incubation in skeletal muscle cells. All treatment groups showed an increase in intracellular calcium levels in comparison with the control group when calcium was present in the buffer. Moreover, the effect of ZnSO_4_ on calcium flux was greater than capsaicin in both calcium-free and calcium-containing buffers. 

### 2.3. Capsaicin and Zinc Phosphorylate CAMKK2 in C2C12 Skeletal Muscle Cells

An increase in cytosolic calcium concentration in skeletal muscle cells leads to phosphorylation of CAMKK2, which upregulates GLUT4 [28]. To examine the effect of capsaicin or ZnSO_4_ + NaPy on the phosphorylation status and activation of CAMKK2, skeletal muscle cells were treated with 100 µM capsaicin or 20 µM ZnSO_4_ + 10 µM NaPy over 60 min (Figure 3). We observed a significant increase in CAMKK2 phosphorylation in skeletal muscle cells after 15, 30, and 60 min of incubation separately with capsaicin and ZnSO_4_ + NaPy.

### 2.4. Capsaicin and Zinc Phosphorylate CREB and TORC1 in C2C12 Skeletal Muscle Cells

The activation of CREB and its co-activator, TORCs (CREB-TORC complex), increase glucose uptake and metabolic efficiency in skeletal muscle cells [33]. Calcium signalling and cAMP signalling pathways activate CREB and TORC1 [30,33]. The activity of CREB and TORC is dependent on their phosphorylation state. Unlike CREB, TORC1 is activated in the dephosphorylated form [30]. To verify the effect of capsaicin or ZnSO_4_ + NaPy on activation of CREB and TORC1, we treated cells with 100 µM capsaicin and 20 µM ZnSO_4_ + 10 µM NaPy over 60 min (Figure 4). As illustrated in Figure 4a,d, 100 µM capsaicin increases activation of CREB and TORC1 after 15- and 30-min incubation times, respectively, in skeletal muscle cells. Similarly, 20 µM ZnSO4 + 10 µM NaPy activates CREB and TORC1 after 15- and 30-min incubation times, respectively, in C2C12 skeletal muscle cells. 

### 2.5. Capsaicin and Zinc Stimulate Junb and Nr4a3 Expression in C2C12 Skeletal Muscle Cells

As mentioned earlier, the activation of the CREB–TORC complex either suppresses or stimulates expression of the downstream target genes involved in glucose metabolism including the transcription factors *Atf3*, *Junb*, and *Nr4a3* [33,37]. qPCR was performed to assess the effect of capsaicin or ZnSO_4_ + NaPy on the mRNA expression of these transcription factors in skeletal muscle cells. As demonstrated in Figure 5a, capsaicin does not affect the expression of the *Atf3* gene in skeletal muscle cells. Figure 5b,c show that *Junb* and *Nr4a3* mRNA expression are elevated in the cells after 15 min of incubation with 100 µM capsaicin. Similar to capsaicin, 20 µM ZnSO_4_ + 10 µM NaPy does not have any effect on the expression of *Atf3* (Figure 5d); however, Figure 5e,f indicate that zinc increases expression of *Junb* and *Nr4a3* after 15 and 30 min of incubation time in skeletal muscle cells, respectively. 

### 2.6. Calcium Activates Signalling Molecules Involved in Glucose Metabolism 

Elevation in cytosolic calcium concentration and its subsequent downstream signalling events lead to an increase in glucose uptake in skeletal muscle cells [26,38]. As mentioned earlier, CAMKK2, CREB, and TORC1 are key signalling molecules involved in glucose metabolism in skeletal muscle cells. To investigate the effect of cytosolic calcium levels elevated by capsaicin and ZnSO_4_ + NaPy, on the activation of CAMKK2, CREB, and TORC1, we inhibited intracellular calcium in the cells using different concentrations of BAPTA-AM (intracellular calcium chelator). We treated skeletal muscle cells with 0–50 µM of BAPTA-AM for 30 min followed by 30 min incubation with 100 µM capsaicin or 20 µM ZnSO_4_ + 10 µM NaPy and measured the phosphorylation status of CAMKK2, CREB, and TORC1. Our results (data obtained from comparing phosphorylation status of each treatment group with the control group (cells with no BAPTA-AM treatment)) demonstrate that 20 and 50 µM of BAPTA-AM block capsaicin-induced phosphorylation of CAMKK2, whereas 50 µM of BAPTA-AM is required to inhibit 20 µM ZnSO_4_ + 10 µM NaPy -induced phosphorylation of this protein in skeletal muscle cells (Figure 6a,i,e,l). Additionally, activation of CREB and TORC1 by capsaicin is diminished after 30 min of incubation with 50 µM BAPTA-AM (Figure 6b,c,j,k). Moreover, 20 and 50 uM of BAPTA-AM inhibited 20 uM ZnSO_4_ + 10 µM NaPy-induced activation of CREB, while 10, 20, and 50 uM of BAPTA-AM were able to block activation of TORC1 (Figure 6f,g,m,n). Accordingly, to confirm the effect of BAPTA-AM on the reduction of cytosolic calcium concentration, we treated cells with 50 µM of BAPTA-AM for 30 min and conducted live-cell imaging. Comparing calcium flux caused by capsaicin or ZnSO_4_ + µM NaPy without BAPTA-AM treatment (data from Figure 1) with BAPTA-AM treated cells indicates that 50 µM of BAPTA-AM reduces the effect of 100 µM capsaicin or 20 µM ZnSO_4_ + 10 µM NaPy on the elevation of cytosolic calcium level in skeletal muscle cells (Figure 7a–d).

### 2.7. CAMKK2 Phosphorylation Is Involved in Calcium-Induced Activation of Signalling Molecules by Capsaicin and Zinc

Calcium/calmodulin binding to CAMKK2 results in its autophosphorylation and subsequent activation of downstream signalling molecules that play an important role in cellular metabolism. CAMKK2 activation also increases the expression and translocation of GLUT4 and glucose uptake in myotubes [38,39]. To evaluate the effect of CAMKK2 on activation of cell signalling molecules induced by capsaicin or ZnSO_4_ + NaPy, skeletal muscle cells were pre-treated with different concentrations of STO-609 (selective CAMKK2 inhibitor) for 30 min followed by 30 min incubation with 100 µM capsaicin or 20 µM ZnSO_4_ + 10 µM NaPy. As indicated in Figure 8, treatment with increasing concentrations of STO-609 (10, 20, and 50 μM) diminishes capsaicin-induced activation of CAMKK2, CREB, and TORC1 (Figure 8a–d,i–k); however, ZnSO_4_ + NaPy-induced phosphorylation of signalling molecules is not affected by 10, 20, and 50 μM STO-609 (Figure 8e–h,l–n), and higher concentrations of STO-609 are required to decrease activation of CAMKK2, CREB, and TORC1 (Figure 9).

### 2.8. Effect of Cytosolic Calcium Level in Glucose Uptake by Capsaicin and Zinc in Skeletal Muscle Cells

As mentioned earlier, an increase in cytosolic calcium levels leads to glucose uptake in skeletal muscle cells [36]. To assess the effect of cytosolic calcium levels increased by capsaicin, ZnSO_4_ + NaPy, and insulin on glucose uptake, C2C12 skeletal muscle cells were pretreated with 50 μM BAPTA-AM for 30 min followed by 60 min incubation with 100 μM capsaicin, 20 µM ZnSO_4_ + 10 µM NaPy, and 10 nM insulin. Figure 10 illustrates that 30 min of incubation with 50 μM BAPTA-AM significantly reduces glucose uptake by capsaicin, ZnSO_4_ + NaPy, and insulin in skeletal muscle cells. This figure also demonstrates that cells treated with 100 μM capsaicin and 20 µM ZnSO_4_ + 10 µM NaPy in the presence of 50 μM BAPTA-AM had no significant difference in glucose uptake in comparison with the control group. Conversely, glucose uptake in 50 μM BAPTA-AM pretreated cells in the presence of 10 nM insulin is significantly higher than in the control group. 

### 2.9. Capsaicin and Zinc Elevate Intracellular cAMP Levels in C2C12 Skeletal Muscle Cells

Elevation of intracellular cAMP levels stimulates cell signalling events that improve glucose metabolism in skeletal muscle cells [33]. To measure the effect of capsaicin and ZnSO_4_ + NaPy on intracellular cAMP levels, we treated cells with 100 μM capsaicin, 20 µM ZnSO_4_ + 10 µM NaPy, and 50 μM forskolin (as a positive control) for 60 min. Our data indicate that 100 μM capsaicin, 20 µM ZnSO_4_ + 10 µM NaPy, and 50 μM forskolin significantly increase intracellular cAMP level after 60 min of treatment (Figure 11). As previously mentioned, CREB, TORC1, and their target genes are regulated through both elevations of cytosolic calcium and cAMP levels, which subsequently lead to an improvement in glucose metabolism [30,33].

## 3. Discussion

To the best of our knowledge, this is the first study to show that capsaicin and zinc, when tested individually, elevate cytosolic calcium and cAMP levels in C2C12 cells. We also demonstrated that these two bioactive food components, the calcium signalling pathway, and its downstream signalling molecules, including CAMKK2, CREB, and TORC1, regulate the expression of target genes and promote calcium-dependent glucose uptake in C2C12 skeletal muscle cells. Moreover, reduction of cytosolic calcium levels by BAPTA-AM deactivated the CAMKK2, CREB, and TORC1 and reduced capsaicin- and zinc-mediated glucose uptake. This clearly shows the significant role of capsaicin- or zinc-induced calcium release in the activation of calcium signalling molecules as well as glucose uptake in C2C12 cells. Finally, we demonstrated that CAMKK inhibition by STO-609 treatment decreases capsaicin- as well as zinc-stimulated activation of CREB and TORC1, which supports the upstream regulatory role of CAMKK2 in the activation of these proteins. Our results suggest calcium as an important target in studying signalling events involved in glucose metabolism.

The crucial role of nutrients and food components in glucose homeostasis, including maintenance of normal blood glucose level, is well-established [40,41]. Chilli peppers, for example, have medical and pharmacological potential in carbohydrate metabolism disorders owing to their capsaicin content [1,42,43]. Capsaicin acts through the activation of TRPV1, a calcium-permeable ion channel [1,44,45]. Many in vitro, animal, and human studies have confirmed the beneficial role of capsaicin in the modulation of glucose metabolism [1,3,7,12,19,46]. Similarly, zinc affects glucose homeostasis through various pathways, activating signalling molecules involved in glucose uptake and insulin action such as protein kinase B (AKT), small heterodimer partner (SHP), and extracellular signal-regulated protein kinases 1 and 2 (ERK1/2), in both healthy and diabetic individuals [2,4,5,8,15]. Previous studies in our laboratory have shown that capsaicin and zinc cause an elevation in glucose uptake in C2C12 cells through the activation of the CAMKK2- 5’ AMP-activated protein kinase (AMPK) pathway and AKT signalling, respectively [3,22]; however, these pathways are distinct [47]. In the present study, for the first time, we suggest calcium signalling and the downstream events as the common pathway between capsaicin and zinc in glucose uptake.

As mentioned earlier, several studies conducted on the mechanism of action of capsaicin and zinc suggest that these two bioactive food components regulate glucose metabolism through distinct pathways [2,3,19,22]. However, it seems that both capsaicin and zinc stimulate calcium signal transduction, which is perhaps a common pathway shared by these food components [23,24,48]. Our data suggest calcium flux, and its downstream signalling events, are joint pathways between capsaicin and zinc in glucose uptake in C2C12 skeletal muscle cells.

Calcium release to the cytosol from the extracellular space or intracellular stores plays a substantial role in insulin-dependent and -independent glucose uptake in skeletal muscle [26,36]. Increased cytosolic calcium levels contribute to the delivery of glucose into the skeletal muscle cells by activating downstream kinases [49]. The sarcoplasmic reticulum (SR) in skeletal muscle cells spontaneously releases calcium to the cytoplasm, which leads to glucose uptake in these cells [50]. Nevertheless, calcium influx and the subsequent glucose disposal induced by external stimuli are significantly higher in the skeletal muscle [51]. In the present study, we showed that 100 μM capsaicin or 20 µM ZnSO_4_ elevate cytoplasmic calcium levels in skeletal muscle cells. We observed an increase in cytosolic calcium concentration even in the cells incubated in calcium-deficient media. It is worth mentioning that capsaicin and zinc increase cytosolic calcium levels via different pathways. Capsaicin elevates intracellular calcium levels by the activation of the TRPV1 channel [1]. On the other hand, extracellular zinc stimulates calcium flux independent of calcium receptors and through the extracellular G coupled zinc sensing receptors [48]. Zinc-sensing receptors trigger activation of G proteins activate phospholipase-C (PLC), followed by the binding of IP3 to its receptor and a subsequent release of calcium from intracellular stores in a few seconds in HT29 cells [48]. Similarly, intracellular zinc is shown to induce calcium flux in cardiac muscle within a few seconds by binding to ryanodine receptor 2 (RyR2) on the sarcoplasmic reticulum [52]. We showed that capsaicin and zinc treatment cause calcium flux in the cells incubated in calcium free buffer. This supports the effectiveness of capsaicin and zinc on the calcium release from intracellular stores.

The effects of a selective calcium chelator, BAPTA-AM, in several studies, have demonstrated the significant role of cytosolic calcium flux in the activation of signalling molecules and, subsequently, glucose uptake in skeletal muscle and adipose tissues [26,53,54]. Accordingly, we performed a glucose uptake test in skeletal muscle cells treated with capsaicin and ZnSO_4_ in the presence or absence of BAPTA-AM to assess the importance of calcium in capsaicin- and zinc-induced glucose uptake. We observed a reduction in glucose uptake in the BAPTA-AM treated group, suggesting a crucial role for calcium in capsaicin- and zinc-mediated glucose uptake in skeletal muscle cells.

An increase in cytosolic calcium levels mediates GLUT4 translocation to the cell membrane and glucose uptake through the activation of calcium-calmodulin-dependent proteins, including CAMKK2, independent of insulin in skeletal muscle [36]. CAMKK2 is a potent regulator of whole-body glucose metabolism, making it a promising therapeutic target for controlling hyperglycemia [55]. We have previously demonstrated that capsaicin treatment causes increased phosphorylation of CAMKK2 through the activation of the TRPV1 channel in mouse skeletal muscle cells [3]. Our present study shows that, like capsaicin, zinc treatment also leads to phosphorylation of CAMKK2 in skeletal muscle cells. Moreover, increasing concentrations of BAPTA-AM reduced capsaicin- and zinc-induced phosphorylation of CAMKK2, further supporting the role of calcium as the upstream regulator of CAMKK2 in skeletal muscle.

CREB and TORC1 are key transcription elements for the maintenance of efficient glucose metabolism, and similarly to CAMKK2, are shown to be regulated by an elevation in intracellular calcium levels [33,56,57]. CREB and TORC1 play a significant role in glucose homeostasis by modulating target genes including Pparg coactivator 1 alpha (*PGC-1alpha)*, interleukin (*IL-6)*, salt inducible kinase 1 (*SIK1)*, *JUNB*, and *NR4A3*, which stimulate mitochondrial biogenesis and improve nutrient uptake as well as metabolism and therefore represent potential therapeutic targets in promoting normal skeletal muscle metabolism in diabetes [33]. Previously, we showed that zinc activates CREB in skeletal muscle cells [2]. In this study for the first time, we demonstrated that capsaicin also activates CREB in skeletal muscle and that both capsaicin and zinc cause activation/dephosphorylation of TORC1 in skeletal muscle cells. Activation of TORC1 leads to the translocation of this protein from the cytoplasm to the nucleus, which consequently regulates gene expression [58]. TORC1 is shown to alleviate hyperglycemia in experimental diabetes [59]. A *Torc1-*knockout study suggests that mutations in the *Torc1* gene are associated with the presence of IR [56]. Similarly, *Creb*-depletion is also observed in the vascular stroma of insulin-resistant and diabetic rodents [60]. Because of the beneficial roles of CREB and TORC1 in the activation of target genes involved in the control of glucose homeostasis and considering the effectiveness of capsaicin and zinc in the regulation of these two transcriptional factors, it seems that capsaicin and zinc could have a potentially positive impact on the control of glucose metabolism by the activation of CREB, TORC1, and the downstream genes in skeletal muscle cells.

CREB can be activated because of either an increase in intracellular calcium or cAMP levels [32,33]. Protein kinases activated by calcium, such as calmodulin-dependent protein kinase (CAMK), activate CREB by phosphorylation at Ser133 [37]. Our calcium and CAMKK2 inhibition studies indicate that calcium and CAMKK2 are upstream stimulators of CREB and are required for capsaicin- or zinc-dependent activation of this transcription factor in skeletal muscle cells. Further, we showed that TORC1 dephosphorylation/ activation is also dependent on intracellular calcium levels and CAMKK2 phosphorylation status in treated cells. These results again emphasise the crucial role of calcium in activating key signalling molecules involved in glucose metabolism in skeletal muscle.

As mentioned earlier, activation of CREB and its co-activators, TORCs, modulate glucose metabolism by regulating the expression of transcriptional targets of CREB and TORCs including *Atf3*, *Junb*, and *Nr4a3* [33,56,61]. There are conflicting data on the role of ATF3 in glucose metabolism [34,62,63]. ATF3 is shown to negatively affect glucose homeostasis through down-regulation of *Glut4* expression and insulin sensitivity reduction in hepatic cells [34,63]. Moreover, a study has demonstrated that overexpression of *Atf3* in the liver and pancreas of transgenic mice is associated with glucose dyshomeostasis. Additionally, ATF3 is shown to bind to the promoter in pancreatic β cells and suppress the transcription of the related genes [34]. In contrast, ATF3 is shown to play a beneficial role by enhancing glucose tolerance in high-fat diet-induced diabetes and pancreatic β cells dysfunction [62]. However, our results showed that neither capsaicin nor zinc affects *Atf3* expression over 120 min of incubation time. This may be because of inadequate incubation time for capsaicin or zinc to regulate the expression of *Atf3* in the cells; however, more studies are needed to determine the impact of capsaicin and zinc in the regulation of *Atf3* in skeletal muscle cells.

*Junb* and *Nr4a3* increase insulin sensitivity, *Glut4* transcription, and promote glucose uptake in skeletal myocytes [33,64]. *Junb* improves glucose metabolism by promoting the maintenance of skeletal muscle mass and hypertrophy [33,64]. Like *Junb*, *Nr4a3* is also suggested to ameliorate glucose homeostasis in skeletal muscle tissue [65]. Our qPCR data indicated that *Junb* and *Nr4a3* expression is elevated in skeletal muscle cells treated with either capsaicin or zinc. Therefore, the positive impact of capsaicin and zinc on glucose metabolism may in part be due to their influence on these master regulator genes.

Due to the significant effect of cAMP signalling on glycaemic control [35], we evaluated the impact of capsaicin and zinc on intracellular cAMP concentrations in skeletal muscle cells. Many studies have shown the effectiveness of capsaicin on intracellular cAMP elevation in adipose tissue and sensory neurons [10,66,67]. In contrast, there is uncertainty surrounding the influence of zinc on intracellular cAMP levels [68,69]. Maywald et al. reported that zinc causes an increase in intracellular cAMP levels by inhibiting cAMP hydrolysis [69]. In contrast, another group demonstrated a suppressing effect of zinc on cAMP synthesis through the abrogation of adenylyl cyclase activity [68]. However, our data revealed a significant rise in intracellular cAMP levels in capsaicin- or zinc-treated cells. This is in line with previous studies on the effect of capsaicin on intracellular cAMP levels [10,66,67]. A comparison between glucose uptake in the cells treated with BAPTA-AM and without BAPTA-AM suggests that the enhancement of cytosolic calcium concentrations has more effect on glucose uptake compared to intracellular cAMP elevation in the skeletal muscle cells treated with capsaicin or zinc. Our data suggest capsaicin and zinc as effective modulators of calcium-induced glucose uptake that could potentially influence glucose metabolism by stimulating calcium flux and triggering downstream signalling events.

Capsaicin and zinc induce their impacts on cellular processes in different ways [3,70]. Capsaicin acts through the activation of the TRPV1 channel, while zinc-sensing receptors are the link between extracellular zinc and zinc-induced signal transduction in the cells [1,71]. Moreover, glucose metabolism regulated by capsaicin, and zinc is distinct, as most of the studies investigated the AMPK pathway activated by capsaicin and AKT pathway stimulated by zinc and their role in glucose homeostasis [1,19,22,72]. In the present study, we demonstrate that both capsaicin and zinc elevate calcium flux in the skeletal muscle, which leads to glucose uptake in these cells. The original findings of this paper provide novel and valuable information on the mechanism of action of capsaicin and zinc in glucose uptake, which accordingly increases our knowledge of glucose metabolism.

Further inhibition studies are required to understand the role of capsaicin and/or zinc-mediated activation of CAMKK2, CREB, and TORC1 as central proteins in calcium signalling in glucose uptake. This can potentially lead to the development of therapeutic targets for carbohydrate metabolism disorders including IR and T2DM by utilising these two bioactive food components.

## 4. Materials and Methods

### 4.1. Reagents and Antibodies

Capsaicin (cat#M2028), forskolin (cat#F3917), BAPTA-AM (cat#A1076), 3-Isobutyl-1-methylxanthine (cat#17018), 4-(3-Butoxy-4-methoxybenzyl) imidazolidin-2-one (cat#B8279), Pyrithione sodium salt (cat# H3261), and ZnSO_4_ (cat#83265) were purchased from Sigma Aldrich, Melbourne, Australia. STO-609 acetate (cat#ab141591) was obtained from Abcam, Australia. Insulin (cat# 12585014) was purchased from Thermo Fisher, Melbourne, Australia. Antibodies pCaMKK2 (Ser511) (cat#12818), CAMKK2 (cat#16810), pCREB (Ser133) (cat# 9198S), CREB (cat# 9197S), pTORC1/CRTC1 (Ser151) (cat#3359), TORC1/CRTC1 (cat#2587), HRP-linked secondary antibodies (Anti-rabbit (cat#7074) and Anti-mouse (cat#7076)), and GAPDH (cat#5174) were purchased from Cell Signaling Technology, Beverly, MA, USA.

### 4.2. Cell Culture

Skeletal muscle (C2C12) cells (Sigma Aldrich, Melbourne, Australia) were cultured in 10% fetal calf serum and 1% Penicillin-Streptomycin-supplemented Dulbecco′s Modified Eagle′s Medium (DMEM) (Sigma Aldrich, Melbourne, Australia) and grown in an optimal condition of 5% CO_2_ at 37 °C. At approximately 70% confluence, skeletal muscle cells were differentiated to myotubes using a medium containing 2% horse serum (Thermo Fisher, Melbourne, Australia) for 72 h. Three hours before starting the various treatments, skeletal muscle cells were exposed to a serum-free medium. Cells were then treated with capsaicin, ZnSO_4_ + NaPy, insulin, forskolin, and different concentrations of BAPTA-AM and STO-609 as per the following procedures outlined below 

### 4.3. Immunoblot Analysis of Protein Expression 

A total of 4 × 105 cells/well were cultured in 6-well plates and differentiated to myotubes for 72 h. Then, the cells were treated with 100 μM capsaicin; 20 μM ZnSO_4_ + 10 μM NaPy; 10 nM insulin; 50 μM forskolin, 10, 20, and 50 μM BAPTA-AM; and 10, 20, 50, 100, 200, and 400 μM STO-609 at different time points and were lysed using RIPA lysis buffer (Thermo Fisher, Melbourne, Australia) in the presence of phosphatase and protease inhibitor cocktail, as per the manufacturer’s instructions. Total cellular protein concentration was measured using a PierceTM BCA Protein Assay Kit as per the manufacturer’s instructions (Thermo Fisher, Melbourne, Australia). A total of 20 µg of protein was loaded onto a 4–15% Mini-PROTEAN^®^ TGX™ Precast Protein Gels (Bio-Rad, Sydney, Australia) and transferred to a PVDF membrane using a semi-dry Turbo Transfer System (Bio-Rad, Sydney, Australia). The membranes were blocked for 120 min in blocking buffer (5% skim milk in TBST buffer (20 mM Tris, 150 mM NaCl, 0.1% Tween 20, pH 7.5)), followed by an overnight incubation at 4 °C with the specific primary antibodies diluted in blocking solution; pCaMKK2 (1:2000), CAMKK2 (1:4000), pCREB (1:2000), CREB (1:4000), pTORC1 (1:2000), TORC1 (1:4000), and GAPDH (1:4000). The membranes were then washed three times with TBST buffer (5 min each time) and were incubated with HRP-conjugated anti-rabbit or mouse secondary antibodies (1:5000) for 120 min. Then the membranes were washed again with TBST as previously described, and protein bands were visualised using SuperSignal™ West Femto Maximum Sensitivity Substrate (ThermoFisher, Melbourne, Australia) and a Fuji LAS-3000 imaging system (Berthold, Australia). Non-phosphorylated protein antibodies and GAPDH were used to normalise phosphorylated protein expression levels.

### 4.4. Calcium Imaging

A total of 4 × 105 cells/well were seeded in 8-well Chamber Slides (ThermoFisher, Melbourne, Australia) and differentiated to myotubes for 72 h. Skeletal muscle cells were treated with 0 or 50 µM BAPTA-AM in the calcium-free Eagle’s Minimum Essential Medium (EMEM) (cat#M2279) (Sigma Aldrich, Melbourne, Australia) or DMEM for 60 min. Calcium flux was measured using the Fluo-4 Calcium Imaging Kit (cat#F10489) (Thermo Fisher, Melbourne, Australia). Accordingly, the cell medium was replaced with phosphate-buffered saline (PBS) containing 100× PowerLoad™ concentrate (Component B) and Fluo-4, AM, 1000× (Component A), and incubated at 37 °C for 15–30 min, followed by 15–30 min at room temperature. Afterwards, the cells were washed with PBS and 1 mL PBS was added to the top of the cells. Then, 100 µM capsaicin and 20 μM ZnSO_4_ + 10 μM NaPy were added to cells, and live-cell imaging was performed (EVOS™ M5000 Imaging System). Standard FITC settings were used to visualise the cytosolic staining of the Fluo-4, AM dye, and the mean fluorescence intensity was measured utilising ImageJ software. 

### 4.5. cAMP Measurement

The cAMP-Glo™ Assay (Promega, Sydney, Australia) was used to measure intracellular cAMP levels in control and treated cells with capsaicin, zinc, and forskolin (positive control for intracellular cAMP elevation). The assay was based on the principle that cAMP stimulates protein kinase A (PKA) holoenzyme activity, decreasing available ATP and leading to decreased light production in a coupled luciferase reaction. Accordingly, the cells were seeded in a white 96-well plate at a density of 1 × 10^4^ cells/well and differentiated to myotubes for 72 h. On the day of the assay, the cells were treated with 100 µM capsaicin, 20 μM ZnSO_4_ + 10 μM NaPy, and 50 µM forskolin in induction buffer (PBS containing 500 µM isobutyl-1-methylxanthine and 100 µM imidazolidone) for 30 min. Intracellular cAMP levels were measured and calculated following the manufacturer’s instructions.

### 4.6. CAMKK2 Inhibition Study

Cells (a total of 3 × 105) were cultured in 12-well plates and differentiated to myotubes for 72 h. Subsequently, 0, 10, 20, 100, 200, and 400 µM of STO-609 (CAMKK2 inhibitor) were added to the cells and incubated for 30 min followed by another 30 min incubation with 100 µM capsaicin and 20 μM ZnSO_4_ + 10 μM NaPy. Western blots were performed for immunoreactivity of pCAMKK2, pCREB, pTORC1, CAMKK2, CREB, TORC1, and GAPDH as previously described. 

### 4.7. RNA Extraction, cDNA Synthesis, and Quantitative Real-Time PCR (qPCR)

Differentiated C2C12 skeletal muscle cells were treated with 100 µM capsaicin and 20 μM ZnSO_4_ + 10 μM NaPy for up to 60 min. The ISOLATE II RNA Mini Kit (Bioline, Sydney, Australia) was used to extract RNA from skeletal muscle cells as per the manufacturer’s instructions. cDNA was synthesised from total RNA using the High-Capacity cDNA Reverse Transcription Kit (Thermo Fisher, Melbourne, Australia). Primers for qPCR amplification were purchased from QIAGEN (Melbourne, Australia) and included: *Atf3* (GeneGlobe ID# QT00098147), *Nr4a3* (GeneGlobe ID# QT00145873), *Junb* (GeneGlobe ID# QT00241892), and *Gapdh* (PPM02946E). The qPCR assays for gene expression analysis were run as three independent experiments and three technical replicates with a final reaction volume of 20 µL, containing 10 µL of SensiFAST SYBR^®^ No-ROX mix (2×) (Bioline, Sydney, Australia). *Gapdh* was used to normalise the mRNA levels of genes, and the 2^−ΔΔCT^ method was applied to measure the relative changes in gene expression [73].

### 4.8. Glucose Uptake Assay

Myoblasts were cultured in a white 96-well plate at a density of 5 × 10^3^. After 24 h, the cells were differentiated to myotubes for 72 h. Twenty-four hours before the glucose uptake assay, the cell medium was replaced with glucose-free EMEM. On the day of the assay, C2C12 skeletal muscle cells were treated with 100 µM capsaicin, 20 μM ZnSO_4_ + 10 μM NaPy, and 10 nM insulin for 60 min in the presence or absence of 50 µM BAPTA-AM. The glucose uptake assay was conducted applying the Glucose Uptake-GloTM Assay Kit (Promega, Sydney, Australia) according to the manufacturer’s instructions. Briefly, 50 µL of 0.1 mM 2DG was added to cells and incubated for 30 min at 24 °C. Then, 25 µL of stop buffer and neutralization buffer were added to each well. Finally, 100 µL of 2DG6P was added to the wells, and the plate was incubated for 60 min at 25 °C. Luminescence was recorded with 0.3–1 s integration on the TECAN infinite M200 PRO plate reader, and glucose uptake level was calculated as per the manufacturer’s instruction. 

### 4.9. Statistical Analysis

The *t* test statistic was used to determine the significant difference between the mean value of two groups (the mean value of only two groups, a treatment group, and control group, was compared throughout this study) by applying GraphPad Prism 8 and expressed as the mean ± Standard Deviation (SD). *p* < 0.05 represents a statistically significant difference between the two groups. 

## 5. Conclusions

To summarise, the present study demonstrates for the first time that capsaicin and zinc treatment promotes glucose uptake through calcium signalling independent of insulin in C2C12 skeletal muscle cells. We showed that capsaicin and zinc activate CAMKK2, CREB, and TORC1 and regulate the expression of *Junb* and *Nr4a3*, which are involved in glucose metabolism. Our data also indicated that increasing concentrations of the calcium chelator BAPTA-AM causes a reduction in activation of signalling proteins and leads to a decrease in glucose uptake in the skeletal muscle treated with capsaicin and zinc. This confirms the crucial role of calcium in capsaicin- and zinc-induced glucose uptake in skeletal muscle. We also observed that CAMKK2 inhibition diminishes activation of CREB and TORC1, which establishes CAMKK2 as the upstream activator of CREB and TORC1 in capsaicin and ZnSO_4_-treated cells. Our results suggest that targeting calcium signalling pathways activated by capsaicin and zinc in skeletal muscle may have beneficial effects on glucose metabolism in carbohydrate metabolism disorders including IR and T2DM.

## Figures and Tables

**Figure 1 ijms-23-02207-f001:**
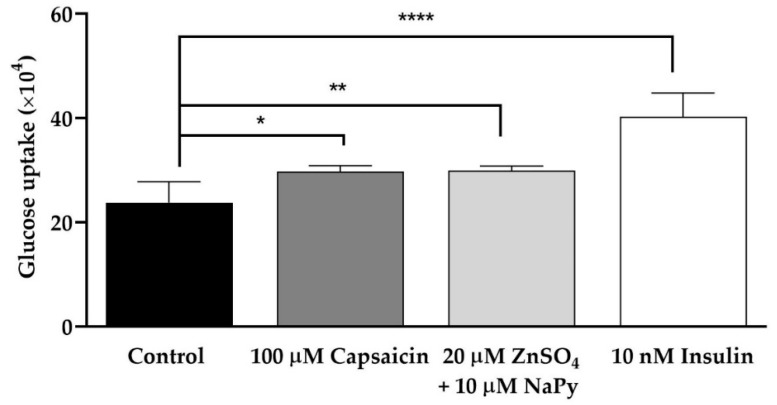
Glucose uptake is induced by capsaicin, ZnSO_4_ + NaPy, and insulin in skeletal muscle cells. Data are presented as mean ± SD of four independent repeats (*n* = 4). * *p* < 0.05, ** *p* < 0.01, and **** *p* < 0.0001 indicate a significant difference between capsaicin, zinc, or insulin-treated and control groups.

**Figure 2 ijms-23-02207-f002:**
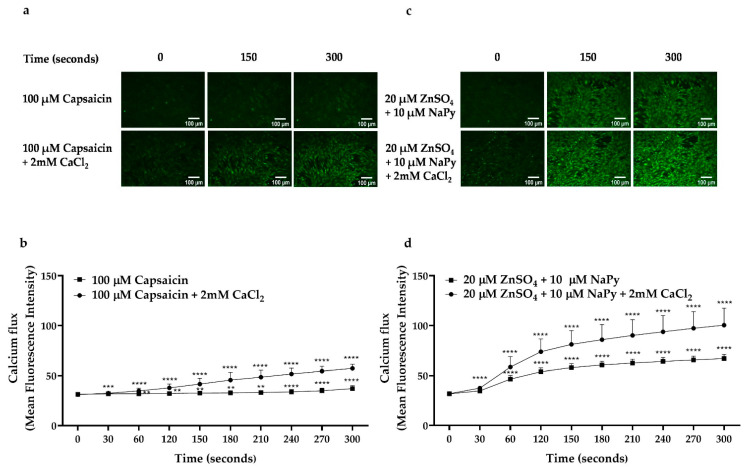
Effect of capsaicin or ZnSO_4_ + NaPy on calcium flux in skeletal muscle cells. (**a**,**b**) Dynamic and representative analysis of an increase in intracellular calcium flux induced by 100 µM capsaicin from extracellular space and intracellular stores in the cells incubated with 2 mM CaCl_2_-containing buffer and from intracellular stores in the cells incubated with calcium-free HBSS buffer over 300 s. (**c**,**d**) Dynamic and representative analysis of the elevation in intracellular calcium flux induced by 20 µM ZnSO_4_ + 10 µM NaPy from extracellular space and intracellular stores in the cells incubated with 2 mM CaCl_2_-containing buffer and from intracellular stores in the cells incubated with calcium-free HBSS buffer with 20 µM ZnSO_4_ + 10 µM NaPy over 300 s. Results are presented as mean ± SD of three independent repeats (*n* = 3). ** *p* < 0.01, *** *p* < 0.001, and **** *p* < 0.0001 indicate a significant difference between capsaicin or zinc treated cells against the control group (i.e., untreated cells at time 0).

**Figure 3 ijms-23-02207-f003:**
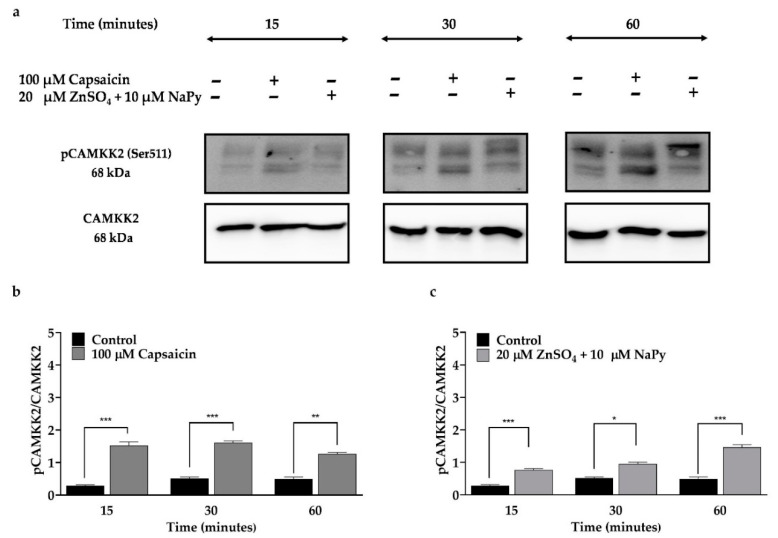
CAMKK2 phosphorylation is separately induced by capsaicin and ZnSO_4_ + NaPy in C2C12 skeletal muscle cells. (**a**) Representative western blot for pCAMKK2 and total CAMKK2 in C2C12 skeletal muscle cells treated with 100 µM capsaicin and 20 µM ZnSO_4_ + 10 µM NaPy over 60 min. (**b**,**c**) Densitometric analysis for pCAMKK2/CAMKK2 in the cells treated with 100 µM capsaicin and 20 µM ZnSO_4_ + 10 µM NaPy over 60 min. Data are presented as mean ± SD of four independent repeats (*n* = 4). * *p* < 0.05, ** *p* < 0.01, and *** *p* < 0.001 indicate a significant difference between capsaicin or zinc treated against control group.

**Figure 4 ijms-23-02207-f004:**
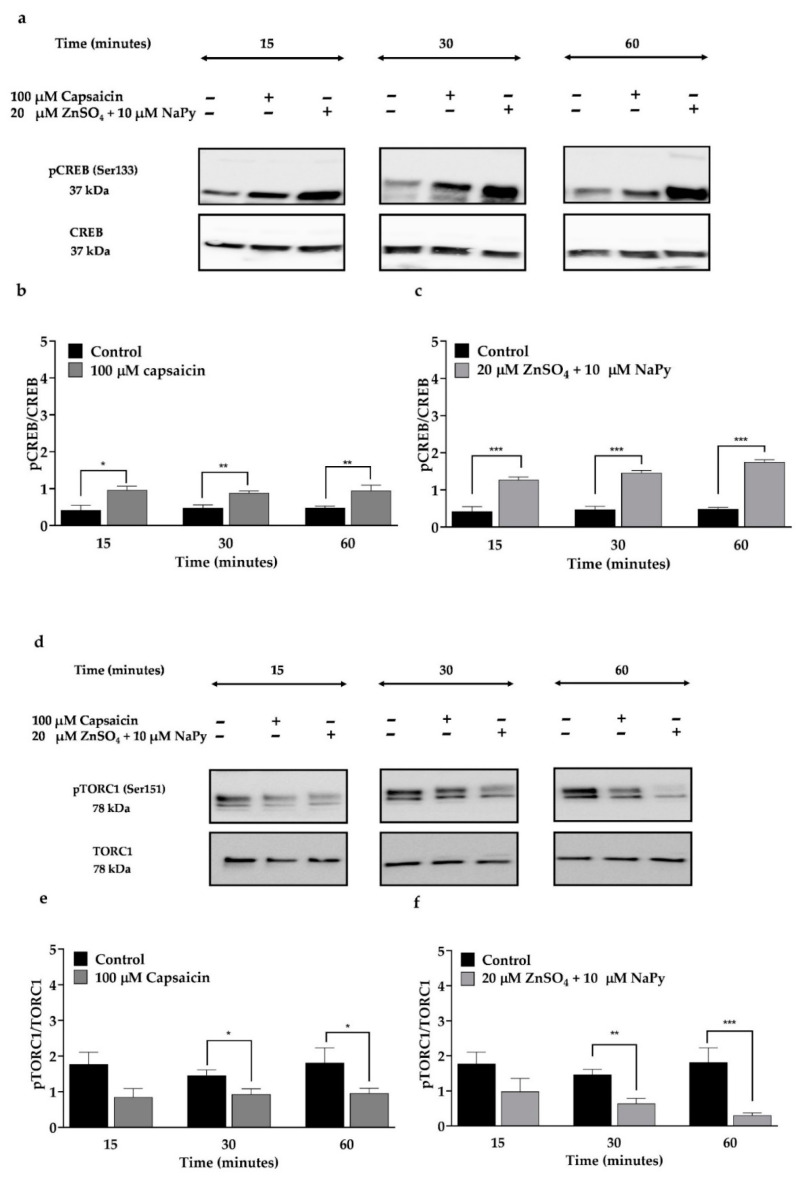
Activation status of CREB and TORC1 by capsaicin or ZnSO_4_ + NaPy in C2C12 skeletal muscle cells. (**a**,**d**) Representative western blots of pCREB, CREB, pTORC1, and TORC1 in the presence and absence of 100 µM capsaicin and 20 µM ZnSO_4_ + 10 µM NaPy over 60 min incubation time. (**b**,**c**) Densitometric analysis of pCREB and CREB in the presence and absence of 100 µM capsaicin and 20 µM ZnSO_4_ + 10 µM NaPy over 60 min incubation time. (**e**,**f**) Densitometric analysis of pTORC1 and TORC1 in the cells treated with 100 µM capsaicin and 20 µM ZnSO_4_ + 10 µM NaPy during 60 min incubation. Data are presented as mean ± SD of four independent repeats (*n* = 4). * *p* < 0.05, ** *p* < 0.01, and *** *p* < 0.001 indicate a significant difference between capsaicin and zinc treated and control groups.

**Figure 5 ijms-23-02207-f005:**
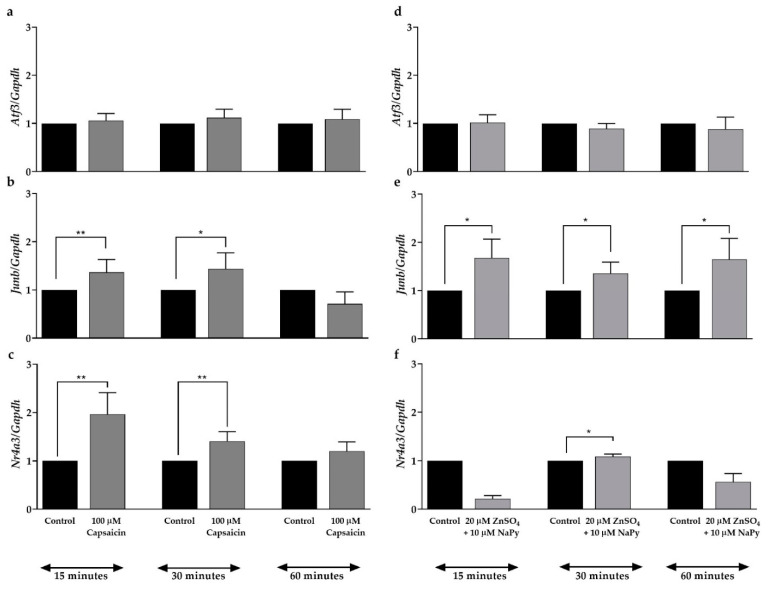
Capsaicin- or ZnSO_4_ + NaPy-induced expression of *Atf3*, *Junb*, and *Nr4a3* over 60 min incubation time. (**a**–**c**) The relative mRNA expression level of *Atf3*, *Junb*, and *Nr4a**3* in control and 100 µM capsaicin groups. (**d**–**f**) The mRNA level of *Atf3*, *Junb*, and *Nr4a3* in 20 µM ZnSO_4_ + 10 µM NaPy-treated and control groups. Data are presented as mean ± SD of three independent repeats (*n* = 3). * *p* < 0.05 and ** *p* < 0.01 demonstrate a significant difference between capsaicin or zinc treated and control groups.

**Figure 6 ijms-23-02207-f006:**
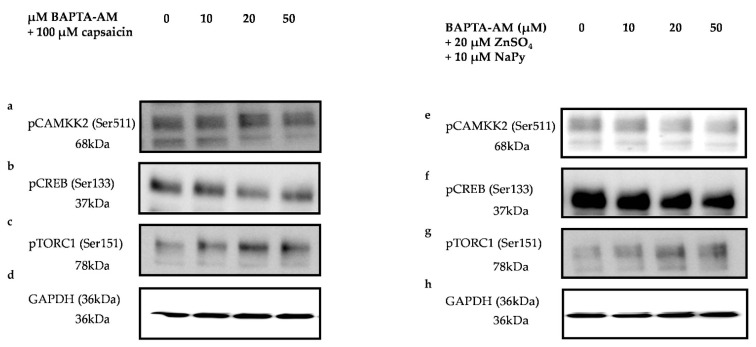
Effect of calcium on the activation of signalling molecules by capsaicin and ZnSO_4_ + NaPy. (**a**–**d**) Immunoblot analysis of pCAMKK2, pCREB, pTORC1, and GAPDH in skeletal muscle cells treated with increasing concentrations of BAPTA-AM (0, 10, 20, and 50 µM) in the presence of 100 µM capsaicin. (**e**–**h**) Western blot results of pCAMKK2, pCREB, pTORC1, and GAPDH in skeletal muscle cells pretreated with increasing concentrations of BAPTA-AM (0, 10, 20, and 50 µM) and with 20 µM ZnSO_4_ + 10 µM NaPy. (**i**–**k**) Densitometric analysis of pCAMKK2, pCREB, and pTORC1 in skeletal muscle cells pretreated with 0, 10, 20, and 50 µM BAPTA-AM in the presence of 100 µM capsaicin. (**l**–**n**) Densitometric analysis of pCAMKK2, pCREB, and pTORC1 in skeletal muscle cells pretreated with 0, 10, 20, and 50 µM BAPTA-AM in the presence of 20 µM ZnSO_4_ + 10 µM NaPy. Data are presented as mean ± SD of three independent repeats (*n* = 3). * *p* < 0.05 and ** *p* < 0.01 indicate a significant difference between capsaicin, zinc, and BAPTA-AM-treated and control (cells with no BAPTA-AM treatment) groups.

**Figure 7 ijms-23-02207-f007:**
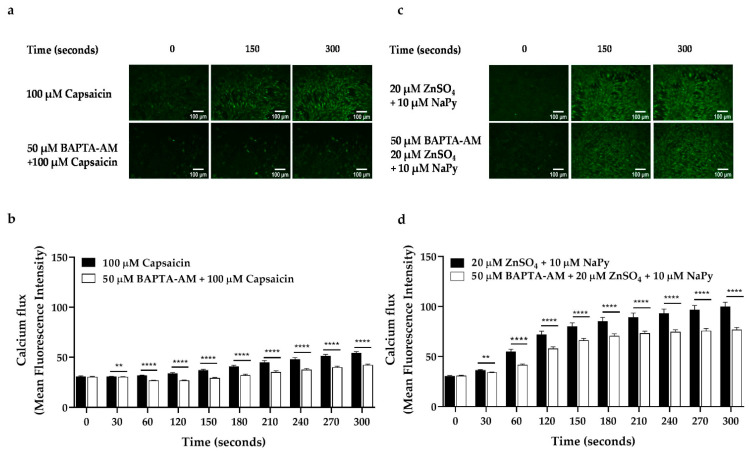
Effect of BAPTA-AM on the elevation of cytosolic calcium level induced by capsaicin and zinc in C2C12 skeletal muscle cells. (**a**,**b**) Dynamic and representative analysis for cytosolic calcium level induced by 100 μM capsaicin in pretreated cells with 50 µM BAPTA-AM for 30 min. (**c**,**d**) Dynamic and representative analysis for cytosolic calcium level induced by 20 µM ZnSO_4_ + 10 µM NaPy in the cells pretreated with 50 µM BAPTA-AM for 30 min. Data are presented as mean ± SD of three independent repeats (*n* = 3). ** *p* < 0.01 and **** *p* < 0.0001 indicate a significant difference between capsaicin or zinc, and BAPTA-AM-treated groups.

**Figure 8 ijms-23-02207-f008:**
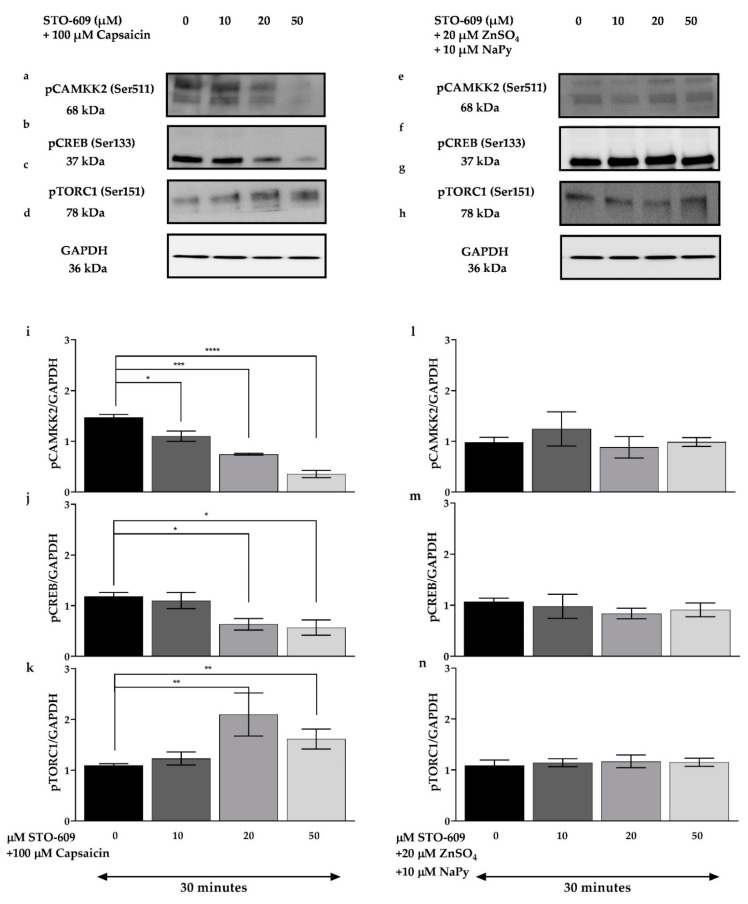
Effect of CAMKK2 inhibition on the phosphorylation status of CAMKK2, CREB, and TORC1 caused by capsaicin and ZnSO_4_ + NaPy in C2C12 skeletal muscle cells. (**a**–**d**) Western blot analysis of pCAMKK2, pCREB, pTORC, and GAPDH in 100 µM capsaicin-treated cells with increasing concentrations of STO-609 (0, 10, 20, and 50 µM). (**e**–**h**) Immunoblot analysis of pCAMKK2, pCREB, and pTORC1 in skeletal muscle cells treated with 20 µM ZnSO_4_ + 10 µM NaPy and increasing concentrations of STO-609. (**i**–**k**) Densitometric analysis of pCAMKK2, pCREB, pTORC1, and GAPDH in pretreated skeletal muscle cells with 0, 10, 20, and 50 µM STO-609 in the presence of 100 µM capsaicin. (**l**–**n**) Densitometric analysis of pCAMKK2, pCREB, pTORC1, and GAPDH in the cells pre-treated with 0, 10, 20, and 50 µM STO-609 in the presence of 20 µM ZnSO_4_ + 10 µM NaPy. Data are presented as mean ± SD of three independent repeats (*n* = 3). * *p* < 0.05, ** *p* < 0.01, *** *p* < 0.001, and **** *p* < 0.0001 indicate a significant difference between capsaicin, zinc, STO-609, and control (cells with no STO-609 treatment) groups.

**Figure 9 ijms-23-02207-f009:**
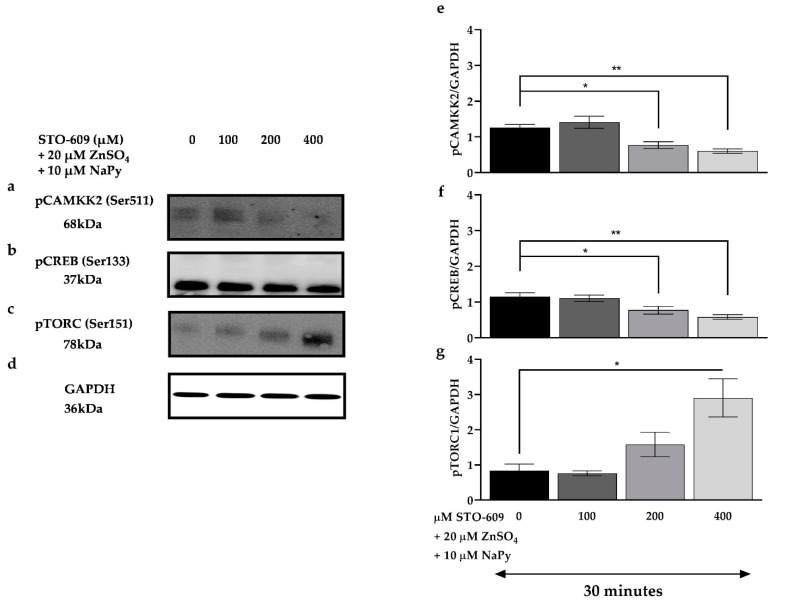
Effect of CAMKK2 inhibition on the phosphorylation status of CAMKK2, CREB, and TORC1 by ZnSO_4_ + NaPy. (**a**–**d**) Western blot analysis of pCAMKK2, pCREB, pTORC, and GAPDH in 20 µM ZnSO_4_ + 10 µM NaPy-treated skeletal muscle cells with increasing concentrations of STO-609 (0, 100, 200, and 400 µM). (**e**–**g**) Densitometric analysis for CAMKK2, CREB, and TORC1 activation by 20 µM ZnSO_4_ + 10 µM NaPy in the cells pretreated with 100, 200, and 400 μM STO-609. Data are presented as mean ± SD of three independent repeats (*n* = 3). * *p* < 0.05 and ** *p* < 0.01indicate a significant difference between zinc, STO-609-treated, and control (cells with no STO-609 treatment) groups.

**Figure 10 ijms-23-02207-f010:**
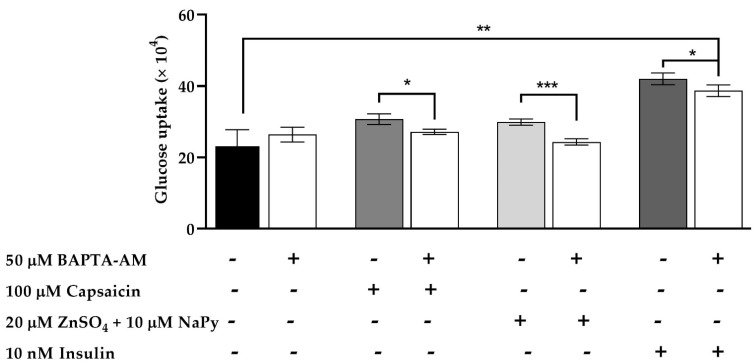
Effect of reduction in cytosolic calcium level by BAPTA-AM in glucose uptake induced by capsaicin, ZnSO_4_ + NaPy, and insulin in skeletal muscle cells. Data are presented as mean ± SD of four independent repeats (*n* = 4). * *p* < 0.05, ** *p* < 0.01, and *** *p* < 0.001 indicate a significant difference between capsaicin, zinc, insulin, and BAPTA-AM-treated and control groups.

**Figure 11 ijms-23-02207-f011:**
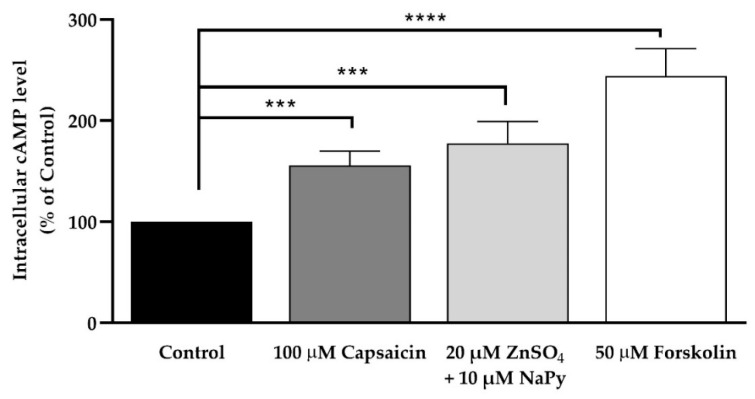
Effect of capsaicin, ZnSO_4_ + NaPy, and forskolin in intracellular cAMP level in skeletal muscle cells. Data are presented as mean ± SD of four independent repeats (*n* = 4). *** *p* < 0.001 and **** *p* < 0.0001 demonstrate a significant difference between capsaicin, zinc, forskolin-treated, and control groups.

## Data Availability

The data presented in this study are available in the manuscript.

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
