# Peer review of "Capsaicin and Zinc Promote Glucose Uptake in C2C12 Skeletal Muscle Cells through a Common Calcium Signalling Pathway"

_ijms, 2022, doi:10.3390/ijms23042207_

Round 1

Reviewer 1 Report

The authors responded in good faith to my comments. The manuscript is improved.

Reviewer 2 Report

The authors have satisfied my critiques

This manuscript is a resubmission of an earlier submission. The following is a list of the peer review reports and author responses from that submission.

Round 1

Reviewer 1 Report

In this paper by Vahidi Ferdowsi et al. entitled "Capsaicin and Zinc Induce Calcium Signalling and Subsequent Glucose Oxidation in Mouse Skeletal Muscle Cells", the authors report a molecular mechanism of glucose metabolism by calcium signaling. They indicate capsaicin and zinc induce the increase of calcium concentration in C2C12 cells. Upregulated calcium signaling activates CaMKK2-CREB-TORC1 signaling. They also indicate that capsaicin and zinc induces glucose oxidation in C2C12 cells. These observations are interesting and may provide a novel therapeutic strategy of metabolic disease. However, I have the following comments:

Major comments:

  1. Much of what the authors present in this study is already identified by their previous studies (Vahidi Ferdowi et al., Cells, 10, 1560, 2021; Vahidi Ferdowi et al., Molecules, 3, 25, 2098, 2020; Norouzi et al., PLoS One, 13, e0191727, 2018). This study has lost novelty. The authors should demonstrate and discuss the significance of their results.

  1. The title is misleading. The authors do not use the skeletal muscle cells derived from mouse in this manuscript. They use C2C12 cells. The authors should reconsider the title.

  1. Fig. 2, 3, 6, 8, 9: I have concerned about the quality of the Western blotting data presented in the manuscript. Vertically sliced blot (even from the same gel) must not be presented as a single blot. A lot of bands are duplicated (100 µM Capsaicin -, 20 µM Zinc – etc.). The authors should consider replacing the Western blots.

  1. Fig. 2, 3, 6, 8, 9: A lot of Western blots are overexposed, making it impossible to use them for quantitative analysis (Fig. 2 CAMKK2, Fig. 6 GAPDH, Fig. 8 pCREB (Ser133) & GAPDH), Fig. 9 pCREB (Ser133) & GAPDH and there are many saturated bands in the gel pictures in the Supplemental data). Over-exposed blots need to be replaced.

  1. Fig. 1, 7: The data is duplicated. The image of “100 µM Capsaicin” and “20 µM Zinc” are duplicated. Fig. 1 and 7 should be combined into one figure.

  1. Line 146: The figure legend is lacking. As there are many errors, the authors should carefully check the manuscript.

  1. Skeletal muscle cells show spontaneous Ca2+ elevation. Why doesn’t spontaneous Ca2+ signaling induce glucose oxidation? The authors should discuss the skeletal muscle-specific Ca2+ signaling and glucose oxidation.

  1. The authors should investigate whether CAMKK2, CREB, TORC1 signaling induce the glucose oxidation in C2C12 cells. This is an important point of this manuscript. The authors should examine the pharmacological experiments (inhibitors and agonist).

  1. Fig. 7 and 10: BAPTA-AM is intracellular calcium chelating drug. It is self-evident that calcium chelating reagents reduce intracellular calcium levels. They should indicate whether capsaicin and zinc directly increase the concentration of intracellular calcium in Fig. 7. In addition, they should indicate whether capsaicin and/or zinc-dependent calcium signaling upregulates the glucose oxidation in C2C12 cells in Fig. 10. The authors should examine the pharmacological experiments by using calcium channel inhibitors.

  1. Why does zinc increase calcium levels in a few minutes? The authors should explain the molecular mechanism.

  1. Methods: What does “20 µM Zinc” mean? What is the source of zinc ion? ZnCl2? ZnSO4? Zn(OH)2? Does the zinc compound change pH? It would be useful to have this information.

Minor comments:

  1. Fig. 1a and c, Fig. 7a and c: Scale bars should be included.

  1. Line 219: Figure 1 should be Figure 7.

  1. Line 263: Figure 8 should be Figure 10.

  1. Line 265: Figure 8 should be Figure 10.

  1. Line 281: Figure 9 should be Figure 11.

  1. Line 284: Figure 11 should be Figure 10.

  1. Line 456: Why the authors use the C2C12 cells 72 hours after differentiation. Are these cells matured?

Author Response

AUTHORS’ RESPONSE TO REVIEWERS’ COMMENTS

Journal: International Journal of Molecular Sciences

Title of the paper: Capsaicin and Zinc Induce Glucose Oxidation in C2C12 Skeletal Muscle Cells through a Common Calcium Signalling Pathway

Corresponding author: Dr. Stephen Myers

Manuscript ID: ijms-1514587

The authors Parisa Vahidi Ferdowsi, Kiran Ahuja, Jeffrey Beckett, and Stephen Myers thank the reviewer 1 for their constructive feedback. We have addressed each of the concerns and comments below and have highlighted the manuscript where appropriate. We believe the manuscript is now stronger and better positioned for publication in the International Journal of Molecular Sciences.

Reviewer 1 comments:

General comments:

In this paper by Vahidi Ferdowsi et al. entitled "Capsaicin and Zinc Induce Calcium Signalling and Subsequent Glucose Oxidation in Mouse Skeletal Muscle Cells", the authors report a molecular mechanism of glucose metabolism by calcium signaling. They indicate capsaicin and zinc induce the increase of calcium concentration in C2C12 cells. Upregulated calcium signaling activates CaMKK2-CREB-TORC1 signaling. They also indicate that capsaicin and zinc induces glucose oxidation in C2C12 cells. These observations are interesting and may provide a novel therapeutic strategy of metabolic disease. However, I have the following comments:

Authors Response                                           

Major Comments:

Comment 1: 1. Much of what the authors present in this study is already identified by their previous studies (Vahidi Ferdowi et al., Cells, 10, 1560, 2021; Vahidi Ferdowi et al., Molecules, 3, 25, 2098, 2020; Norouzi et al., PLoS One, 13, e0191727, 2018). This study has lost novelty. The authors should demonstrate and discuss the significance of their results.

We have revised the manuscript to clearly present the novelty of our research and the significance of this study (lines: 283, 288-296, 305-313, and 427-434 highlighted in yellow).

Comment 2:  The title is misleading. The authors do not use the skeletal muscle cells derived from mouse in this manuscript. They use C2C12 cells. The authors should reconsider the title.

The title has been revised to include the C2C12 cell line.

Comment 3: Fig. 2, 3, 6, 8, 9: I have concerned about the quality of the Western blotting data presented in the manuscript. Vertically sliced blot (even from the same gel) must not be presented as a single blot. A lot of bands are duplicated (100 µM Capsaicin -, 20 µM Zinc – etc.). The authors should consider replacing the Western blots.

We decided to separate the “control/ capsaicin-treatment” and “control/zinc-treatment” groups from the same gel to make it easier for the readers to follow the manuscript. This presentation of gels is common in the field. Below are papers published in reputable journals  that have applied the same method for their western blot data presentation:

  1. Pilipow, K., Basso, V., Migone, N. and Mondino, A., 2019. Correction: Monoallelic Germline TSC1 Mutations Are Permissive for T Lymphocyte Development and Homeostasis in Tuberous Sclerosis Complex Individuals. PloS one14(6), p.e0218354.
  2. Martelli, M.P., Manes, N., Liso, A., Pettirossi, V., Galletti, B.V., Bigerna, B., Pucciarini, A., De Marco, M.F., Pallotta, M.T., Bolli, N. and Sborgia, M., 2008. A western blot assay for detecting mutant nucleophosmin (NPM1) proteins in acute myeloid leukaemia. Leukemia22(12), pp.2285-2288.
  3. Nashat, A.H. and Langer, R., 2003. Temporal characteristics of activation, deactivation, and restimulation of signal transduction following depolarization in the pheochromocytoma cell line PC12. Molecular and cellular biology23(14), pp.4788-4795.
  4. Zhao, D.Y., Gish, G., Braunschweig, U., Li, Y., Ni, Z., Schmitges, F.W., Zhong, G., Liu, K., Li, W., Moffat, J. and Vedadi, M., 2016. SMN and symmetric arginine dimethylation of RNA polymerase II C-terminal domain control termination. Nature529(7584), pp.48-72.
  5. Mikhailov, V., Mikhailova, M., Pulkrabek, D.J., Dong, Z., Venkatachalam, M.A. and Saikumar, P., 2001. Bcl-2 prevents Bax oligomerization in the mitochondrial outer membrane. Journal of Biological Chemistry276(21), pp.18361-18374.

As the reviewer correctly mentioned, pCAMKK2, pCREB, and pTORC1 western blot bands are duplicate; however, it is standard to have duplicate bands for pCAMKK2 and pTORC1 according to the Cell Signaling Technology (CST) and Sigma-Aldrich (the primary antibodies providers). Please see the related information in the below links:

  • pCAMKK2 (1): https://www.cellsignal.com/products/primary-antibodies/phospho-camkk2-ser511-antibody/12818
  • pCAMKK2 (2): https://www.cellsignal.com/products/primary-antibodies/phospho-camkk2-ser495-antibody/16737
  • pCRTC1: https://www.sigmaaldrich.com/AU/en/product/mm/abe560

Regarding the pCREB western blots, we amended the manuscript based on the reviewer’s comment by removing the pATF1 bands from pCREB western blots in Figure 3a and c, Figure 6b and f, Figure 8b and f, and Figure 9b and reanalysed results again.

Comment 4: Fig. 2, 3, 6, 8, 9: A lot of Western blots are overexposed, making it impossible to use them for quantitative analysis (Fig. 2 CAMKK2, Fig. 6 GAPDH, Fig. 8 pCREB (Ser133) & GAPDH), Fig. 9 pCREB (Ser133) & GAPDH and there are many saturated bands in the gel pictures in the Supplemental data). Over-exposed blots need to be replaced.

We replaced Figure 2a and c, Figure 6d and h, Figure 8b, f, d, and h, and Figure 9b and d, reanalysed the relevant data and changed the densitometry graphs where it was applicable.

Comment 5: The data is duplicated. The image of “100 µM Capsaicin” and “20 µM Zinc” are duplicated. Fig. 1 and 7 should be combined into one figure.

This study contains two different parts: in the first part, we show the effect of capsaicin and zinc on calcium signalling and glucose oxidation. The second section is about investigating the effect of calcium inhibition on the subsequent signalling events as well as glucose oxidation. The second part of the paper begins from Figure 7, where we demonstrate the robustness of our study by confirming the inhibitory effect of BAPTA-AM on intracellular calcium levels. Although Figure 7 contains some data from figure 1, there are new and different results from Figure 1 in Figure 7. In Figure 1, we demonstrate that capsaicin and zinc increase calcium flux in the cells, while in Figure 7 we show the inhibitory effect of BAPTA-AM on capsaicin- or zinc-induced calcium flux by using some data from Figure 1 (increased cytosolic calcium levels data caused by capsaicin or zinc in the absence of BAPTA-AM).

We believe that keeping these two figures separate helps the manuscript to maintain its flow and makes it easier for readers to follow the study. Nevertheless, to avoid confusion about the data presented in Figures 1 and 7, we have amended the manuscript (lines: 199-201 highlighted in yellow).

Comment 6: Line 146: The figure legend is lacking. As there are many errors, the authors should carefully check the manuscript.

Figure 3 includes two different parts (a-d and e-h). We added these two figures separately to the manuscript to maintain the quality of Figure 3. This is also applicable for Figure 6 (a-h and i-n) and Figure 8 (a-h and i-n). We removed lines between two parts of these figures to avoid confusion. Based on the reviewer’s request we double-checked legends and revised them where it was required.

Comment 7: Skeletal muscle cells show spontaneous Ca2+ elevation. Why doesn’t spontaneous Ca2+ signaling induce glucose oxidation? The authors should discuss the skeletal muscle-specific Ca2+ signaling and glucose oxidation.

We have revised the manuscript with the following text (lines: 323-327 and 332-336 and highlighted in yellow).

Comment 8: The authors should investigate whether CAMKK2, CREB, TORC1 signaling induce the glucose oxidation in C2C12 cells. This is an important point of this manuscript. The authors should examine the pharmacological experiments (inhibitors and agonist).

Understanding the mechanism of action of potential therapeutic agents requires systematic step-by-step inquiry. In our case, the first step was to test whether capsaicin and zinc - two natural compounds- have any impact on glucose oxidation through the calcium signalling pathway (which we hypothesised as the common pathway between capsaicin and zinc). If so, then which proteins are activated in response to the elevation of intracellular calcium levels and calcium signalling? Now that we have found that calcium signalling is the common pathway between capsaicin and zinc in glucose oxidation and we investigated the signalling proteins (CAMKK2, CREB, and TORC1) involved in this pathway, the next step will be to extend the work by conducting more pharmacological experiments to show the effect of these signalling proteins in glucose oxidation. We have added a sentence on the future of this research in the manuscript (lines: 430-434 highlighted in yellow).

Comment 9: Fig. 7 and 10: BAPTA-AM is intracellular calcium chelating drug. It is self-evident that calcium chelating reagents reduce intracellular calcium levels. They should indicate whether capsaicin and zinc directly increase the concentration of intracellular calcium in Fig. 7. In addition, they should indicate whether capsaicin and/or zinc-dependent calcium signaling upregulates the glucose oxidation in C2C12 cells in Fig. 10. The authors should examine the pharmacological experiments by using calcium channel inhibitors.

As the reviewer correctly mentioned, BAPTA-AM is a cell membrane permeable drug and reduces intracellular calcium levels; however, by having control groups for each experiment we indicated the direct effect of capsaicin and zinc on the elevation of intracellular calcium levels as well as the activation of signalling molecules. For example, comparing time 0 (control group) with the next time frames (in which cells were treated with capsaicin or zinc with or without BAPTA-AM) in Figures 1 and 7 clearly indicates the direct effect of capsaicin or zinc on the elevation of intracellular calcium levels within a timeframe of 300 seconds.

Comparing glucose oxidation rate between cells with no BAPTA-AM treatments and those which are treated with BAPTA-AM in Figure 10 indicates that elevated calcium levels by capsaicin or zinc significantly affect glucose oxidation stimulated by these two nutrients in C2C12 cells.  Moreover, Figure 6 demonstrates that BAPTA-AM significantly reduces activation of signalling molecules regulated by capsaicin or zinc which supports that calcium acts as an upstream activator of CAMKK2, CREB, and TORC1 in the cells treated with capsaicin or zinc. From the above information, the present study (with no calcium channel inhibition studies) shows that capsaicin- or zinc-induced calcium signalling upregulates glucose oxidation in C2C12 cells. However, in the future, we will consider further investigation by performing the suggested calcium channel inhibition studies.

To clarify the effect of capsaicin and zinc on intracellular calcium elevation, activation of signalling molecules, and glucose oxidation, we amended the manuscript (lines: 288-296 highlighted in yellow).

Comment 10: Why does zinc increase calcium levels in a few minutes? The authors should explain the molecular mechanism.

In the experiments conducted in this study, zinc includes 10 μM pyrithione sodium and 20 μM ZnSO4. We added pyrithione sodium to ZnSO4 solution to permeabilise skeletal muscle cells to zinc. Below are the relevant articles that utilise this procedure:

  • Taylor, K.M., Hiscox, S., Nicholson, R.I., Hogstrand, C. and Kille, P., 2012. Protein kinase CK2 triggers cytosolic zinc signaling pathways by phosphorylation of zinc channel ZIP7. Science signaling, 5(210), pp.ra11-ra11.
  • Taylor, K.M., Vichova, P., Jordan, N., Hiscox, S., Hendley, R. and Nicholson, R.I., 2008. ZIP7-mediated intracellular zinc transport contributes to aberrant growth factor signaling in antihormone-resistant breast cancer cells. Endocrinology, 149(10), pp.4912-4920.
  • Krenn, B.M., Gaudernak, E., Holzer, B., Lanke, K., Van Kuppeveld, F.J.M. and Seipelt, J., 2009. Antiviral activity of the zinc ionophores pyrithione and hinokitiol against picornavirus infections. Journal of virology, 83(1), pp.58-64.

As above mentioned, pyrithione sodium facilitates zinc transport across the cell membrane rapidly. Then, zinc activates the epidermal growth factor receptor which leads to a transient increase in the cytosolic concentration from intracellular stores and calcium influx from the extracellular medium. Below are some relevant articles:

  • Taylor, K.M., Vichova, P., Jordan, N., Hiscox, S., Hendley, R. and Nicholson, R.I., 2008. ZIP7-mediated intracellular zinc transport contributes to aberrant growth factor signaling in antihormone-resistant breast cancer cells. Endocrinology, 149(10), pp.4912-4920.
  • Villalobo, A., Ruano, M.J., Palomo-Jiménez, P.I., Li, H. and Martín-Nieto, J., 2000. The epidermal growth factor receptor and the calcium signal. Calcium: The molecular basis of calcium action in biology and medicine, pp.287-303.

We have amended the manuscript by adding the molecular mechanism of capsaicin and zinc in cytosolic calcium elevation (lines: 332-336 highlighted in yellow).

Comment 11: Methods: What does “20 µM Zinc” mean? What is the source of zinc ion? ZnCl2? ZnSO4? Zn(OH)2? Does the zinc compound change pH? It would be useful to have this information.

In this study 20 µM includes: 20 µM ZnSO4 + 10 µM pyrithione sodium salt. Accordingly, we have revised the manuscript (lines: 439 and 459 highlighted in yellow).

Minor Comments:

Minor comment 1: Fig. 1a and c, Fig. 7a and c: Scale bars should be included.

Scales have been added to Figures 1a and c, and 7a and c.

Minor comment 2: Line 219: Figure 1 should be Figure 7.

Please refer to the authors response to comment 5.

Minor comment 3: Line 263: Figure 8 should be Figure 10.

Please see the authors responses to the comments 5 and 6.

Minor comment 4: Line 265: Figure 8 should be Figure 10.

Please see the authors responses to the comments 5 and 6.

Minor comment 5: Line 281: Figure 9 should be Figure 11.

Please refer to the authors responses to the comments 5 and 6.

Minor comment 6: Line 284: Figure 11 should be Figure 10.

Please see the authors responses to the comments 5 and 6.

Comment 7: Line 456: Why the authors use the C2C12 cells 72 hours after differentiation. Are these cells matured?

Several studies confirm that myoblasts differentiate to myotubes after 72 hours of incubation time with 2% horse serum. Below are some examples:

  • Zhang, H., Wen, J., Bigot, A., Chen, J., Shang, R., Mouly, V. and Bi, P., 2020. Human myotube formation is determined by MyoD–Myomixer/Myomaker axis. Science Advances, 6(51), p.eabc4062.
  • Nowak, S.J., Nahirney, P.C., Hadjantonakis, A.K. and Baylies, M.K., 2009. Nap1-mediated actin remodeling is essential for mammalian myoblast fusion. Journal of cell science, 122(18), pp.3282-3293.
  • Zhang, H., Shang, R. and Bi, P., 2021. Feedback regulation of Notch signaling and myogenesis connected by MyoD–Dll1 axis. PLoS Genetics, 17(8), p.e1009729.

Moreover, in the previous research conducted in our lab, we demonstrated that C2C12 cells differentiate to myotubes after 72 hours of incubation time with 2% horse serum contained DMEM. Below, please find the mentioned manuscript:

Vahidi Ferdowsi, P., Ahuja, K.D., Beckett, J.M. and Myers, S., 2021. TRPV1 Activation by Capsaicin Mediates Glucose Oxidation and ATP Production Independent of Insulin Signalling in Mouse Skeletal Muscle Cells. Cells, 10(6), p.1560.

Reviewer 2 Report

Pg 2 lines 57-58.  The statement “This shows that zinc deficiency is associated with the development of IR and T2DM 57 mainly through the elevation of oxidative stress and inflammatory factors in the patients 58 [9,16].” Is not a logical conclusion from the data presented, in that oxidative stress and inflammatory factors were not discussed in the paragraph.  This is a minor point, because the citations given (9, 16) support the statement.  However, it reads a little odd since it is stated as a conclusion to the paragraph.

Pg 2&3.  Results section 2.1 and 2.2 have the same title.  Please revise title 2.2 to better reflect the content of the section.

Pg 4. Figure 2.  The Western blot data looks convincing, but could use a little more in depth explanation.  Specifically, the phosphorylated CamKK2 has multiple bands and the top most band seems to behave differently in the Capsacin versus zinc experiments. The middle (doublet?) bands seem to show a consistent increase in all time points for the Capsaicin treatments relative to controls, however, in the zinc experiment the last replicate is not so convincing.  The bottom band may or may not change, it is not as obvious, especially for zinc. Please include some arrows or description in the legend to highlight which bands are relevant. Also, there is no independent loading control, which is extremely important in the interpretation of this data, since the metric is a ratio of phosphorylated to unphosphorylated protein.   If the unphosphorylated protein levels change between conditions the ratio will not give a true representation of the data.

Pg 5 figure 3.  The Western blot data needs independent loading controls.  The gel pictures are cut and pasted from different gels suggesting that the experiments were done at different times.  While this is not a major concern since the authors are comparing control to experimental only within time points, it is always better to have the comparisons run in a single gel. Also some of the images are not well aligned giving the impression that the bands are slightly shifted.

Fig 6. The data for CamKK2 does not seem to correlate well with the graph.  This is a minor point since it is clearly different from the effect in the absence of BAPTA, however, if you have an image that is a better representative of the data shown in the graph it might be more appropriate.  Also, the significance bars in j, K, and L are hard to interpret.  Do they indicate everything is different from each other?  Perhaps a better method of indicating differences can add clarity.

Figure 7 is mislabeled fig 1.

At fig 9. Panel e and f.  Once again, the bar representing significant differences is hard to interpret. All of the conditions do not look significantly different.  Also in panel g, are the 0 and 100 uM conditions significantly different?  Please clarify.

2.8. Effect of Cytosolic Calcium Level in Glucose Oxidation by Capsaicin and Zinc in Skeletal 257 Muscle Cells. This section is confusing as it seems to refer to the wrong figures in the text.  Line 264-265 refers to glucose oxidation in figure 8.  Figure 8 doesn’t mention glucose oxidation and it seems the text should refer to figure 10.

Figure 10.  The bars representing significant differences are confusing (same comment as above)

Line 281-282 should refer to figure 11?

Line 283-285 is completely redundant with line 281-282.  Please remove.

The discussion is too long and should be simplified.

Statistics:  A T-test is inappropriate for comparison when more than two groups are compared in the analysis.  These results need to be analyzed with ANOVA.

Author Response

AUTHORS’ RESPONSE TO REVIEWERS’ COMMENTS

Journal: International Journal of Molecular Sciences

Title of the paper: Capsaicin and Zinc Induce Glucose Oxidation in C2C12 Skeletal Muscle Cells through a Common Calcium Signalling Pathway

Corresponding author: Dr. Stephen Myers

Manuscript ID: ijms-1514587

The authors Parisa Vahidi Ferdowsi, Kiran Ahuja, Jeffrey Beckett, and Stephen Myers thank the reviewer 2 for their constructive feedback. We have addressed each of the concerns and comments below and have highlighted the manuscript where appropriate. We believe the manuscript is now stronger and better positioned for publication in the International Journal of Molecular Sciences.

Reviewer 2 comments:

Comments and Suggestions for Authors

Comment 1: Pg 2 lines 57-58.  The statement “This shows that zinc deficiency is associated with the development of IR and T2DM 57 mainly through the elevation of oxidative stress and inflammatory factors in the patients 58 [9,16].” Is not a logical conclusion from the data presented, in that oxidative stress and inflammatory factors were not discussed in the paragraph.  This is a minor point, because the citations given (9, 16) support the statement.  However, it reads a little odd since it is stated as a conclusion to the paragraph.

The statement has been removed from the manuscript based on the reviewer’s comment (please see lines: 50-52 highlighted in yellow).

Comment 2: Pg 2&3.  Results section 2.1 and 2.2 have the same title.  Please revise title 2.2 to better reflect the content of the section.

The title of Results section 2.2 has been changed to “Capsaicin and Zinc Phosphorylate CAMKK2 in C2C12 Skeletal Muscle Cells” and highlighted in yellow.

Comment 3: Figure 2.  The Western blot data looks convincing, but could use a little more in depth explanation.  Specifically, the phosphorylated CamKK2 has multiple bands and the top most band seems to behave differently in the Capsacin versus zinc experiments. The middle (doublet?) bands seem to show a consistent increase in all time points for the Capsaicin treatments relative to controls, however, in the zinc experiment the last replicate is not so convincing.  The bottom band may or may not change, it is not as obvious, especially for zinc. Please include some arrows or description in the legend to highlight which bands are relevant. Also, there is no independent loading control, which is extremely important in the interpretation of this data, since the metric is a ratio of phosphorylated to unphosphorylated protein.   If the unphosphorylated protein levels change between conditions the ratio will not give a true representation of the data.

According to the manufacturer (Cell Signaling) examples of pCAMKK2 western blot analysis (please see the below links), it is standard to get more than one band for the pCAMKK2 with ≈ 68 kDa molecular weight. As a result, we considered all the bands with ≈ 68 kDa molecular weight as pCAMKK2 for the densitometric analysis, and based on this we discussed the effect of capsaicin or zinc on the phosphorylation status of this protein. 

  • https://www.cellsignal.com/products/primary-antibodies/phospho-camkk2-ser511-antibody/12818 
  • https://www.cellsignal.com/products/primary-antibodies/phospho-camkk2-ser495-antibody/16737

As the reviewer correctly mentioned, in general, it is important to have loading control for the western blot analysis; however, the normalisation of a phosphorylated protein to its total expression also allows the ratio of phosphorylated proteins to be assessed (please see the below links). In Figures 2 and 3 we aimed to measure the ratio of phosphorylated to unphosphorylated CAMKK2, CREB, and TORC1 to find the effect of capsaicin or zinc on the activation of these proteins. Also, the total (unphosphorylated) proteins are at the same level during the experiments presented in this study.

  • https://www.bio-rad-antibodies.com/tips-western-blot-detection-of-phosphorylation-events.html?JSESSIONID_STERLING=7894073E72F64FDC3A0C0148CD648D7E.ecommerce2&&evCntryLang=AU-enthirdPartyCookieEnabled=false#2
  • https://www.rndsystems.com/resources/articles/methods-detecting-protein-phosphorylation
  • Bass, J.J., Wilkinson, D.J., Rankin, D., Phillips, B.E., Szewczyk, N.J., Smith, K. and Atherton, P.J., 2017. An overview of technical considerations for Western blotting applications to physiological research. Scandinavian journal of medicine & science in sports, 27(1), pp.4-25.

Comment 4: Pg 5 figure 3.  The Western blot data needs independent loading controls.  The gel pictures are cut and pasted from different gels suggesting that the experiments were done at different times.  While this is not a major concern since the authors are comparing control to experimental only within time points, it is always better to have the comparisons run in a single gel. Also some of the images are not well aligned giving the impression that the bands are slightly shifted.

Please refer to our response to comment 3 about the need for independent loading control for western blot data. As the reviewer correctly mentioned, Figure 3 contains cut and pasted blots; however, all the blots are from the same gel. We separated blots as “control/ capsaicin-treatment” and “control/zinc-treatment” groups to make the manuscript easier to follow for the readers. This sort of western blot results presentation is common in the field. Please find below some of the papers published in reputable journals in the field that applied the same method for their western blot data presentation:

  1. Pilipow, K., Basso, V., Migone, N. and Mondino, A., 2019. Correction: Monoallelic Germline TSC1 Mutations Are Permissive for T Lymphocyte Development and Homeostasis in Tuberous Sclerosis Complex Individuals. PloS one, 14(6), p.e0218354.
  2. Martelli, M.P., Manes, N., Liso, A., Pettirossi, V., Galletti, B.V., Bigerna, B., Pucciarini, A., De Marco, M.F., Pallotta, M.T., Bolli, N. and Sborgia, M., 2008. A western blot assay for detecting mutant nucleophosmin (NPM1) proteins in acute myeloid leukaemia. Leukemia, 22(12), pp.2285-2288.
  3. Nashat, A.H. and Langer, R., 2003. Temporal characteristics of activation, deactivation, and restimulation of signal transduction following depolarization in the pheochromocytoma cell line PC12. Molecular and cellular biology, 23(14), pp.4788-4795.
  4. Zhao, D.Y., Gish, G., Braunschweig, U., Li, Y., Ni, Z., Schmitges, F.W., Zhong, G., Liu, K., Li, W., Moffat, J. and Vedadi, M., 2016. SMN and symmetric arginine dimethylation of RNA polymerase II C-terminal domain control termination. Nature, 529(7584), pp.48-72.
  5. Mikhailov, V., Mikhailova, M., Pulkrabek, D.J., Dong, Z., Venkatachalam, M.A. and Saikumar, P., 2001. Bcl-2 prevents Bax oligomerization in the mitochondrial outer membrane. Journal of Biological Chemistry, 276(21), pp.18361-18374.

Figures 2, 3, 6, 8, and 9 have been revised based on the reviewer’s comment.

Comment 5: Fig 6. The data for CamKK2 does not seem to correlate well with the graph.  This is a minor point since it is clearly different from the effect in the absence of BAPTA, however, if you have an image that is a better representative of the data shown in the graph it might be more appropriate.  Also, the significance bars in j, K, and L are hard to interpret.  Do they indicate everything is different from each other?  Perhaps a better method of indicating differences can add clarity.

As the reviewer correctly mentioned, the western blot image presented in Figure 6a cannot clearly demonstrate changes in CAMKK2 phosphorylation with the elevation of BAPTA-AM concentration. Unfortunately, we do not have other blots which clearly illustrate the significant changes observed in the analyses however, the densitometric analysis performed from different repeats of this experiment shows a significant reduction in CAMKK2 phosphorylation rate in BAPTA-AM treated cells compared with the control group.

Based on the reviewer’s request, all the significance lines in the graphs in Figures 2, 3, 4, 5, 6, 8, 9, 10, and 11 have been changed. We believe this way of significance bars presentation makes graphs easier to analyse and interpret by the readers.

Comment 6: Figure 7 is mislabeled fig 1.

The legend of Figure 7 has been revised.

Comment 7: At fig 9. Panel e and f.  Once again, the bar representing significant differences is hard to interpret. All of the conditions do not look significantly different.  Also in panel g, are the 0 and 100 uM conditions significantly different?  Please clarify.

We changed all the significance lines of the graphs to prevent such confusion for the reviewers and readers. Please see Figures 2, 3, 4, 5, 6, 8, 9, and 10.

Data from Figure 9 e and f demonstrate that 400 μM of STO-609 inhibits capsaicin- or zinc-induced CAMKK2 and CREB phosphorylation after 30 minutes of incubation time in mouse skeletal muscle cells. Panel g of this figure also shows that 200 and 400 μM of STO-609 inactivate TORC1 in skeletal muscle cells.

Comment 8: 2.8. Effect of Cytosolic Calcium Level in Glucose Oxidation by Capsaicin and Zinc in Skeletal 257 Muscle Cells. This section is confusing as it seems to refer to the wrong figures in the text.  Line 264-265 refers to glucose oxidation in figure 8.  Figure 8 doesn’t mention glucose oxidation and it seems the text should refer to figure 10.

We have revised the manuscript (please see line 257 highlighted in yellow).

Comment 9: The bars representing significant differences are confusing (same comment as above)

We have changed all the significance lines in the graphs in Figures 2, 3, 4, 5, 6, 8, 9, and 10.

Comment 10: Line 281-282 should refer to figure 11?

We revised the manuscript (please see line 271 highlighted in yellow).

Comment 11: Line 283-285 is completely redundant with line 281-282.  Please remove.

The lines 283-285 have been removed based on the reviewer’s comment.

Comment 12: The discussion is too long and should be simplified.

We have removed some repetitive and unnecessary parts of the discussion.

Comment 13: Statistics:  A T-test is inappropriate for comparison when more than two groups are compared in the analysis.  These results need to be analyzed with ANOVA.

We utilised t tests for the statistical analysis as we only compared the means of two groups: the control group (untreated cells) with a group of treated cells (across only two groups). In this study, we did not compare across the 3 groups or more. For example: in Figure 2, the control group (untreated cells) and sample group (capsaicin or zinc treated cells) were compared based on the mean value of pCAMKK2/CAMKK2. To make it more clear that we compare only two groups throughout this study we changed the significance lines in the graphs in Figures 2, 3, 4, 5, 6, 8, 9, 10, and 11. We have revised the manuscript with the following text (please see lines: 189 and 190, 539-541 highlighted in yellow).

Round 2

Reviewer 1 Report

Although the authors responded to the reviewer's comments, I have the following comments:

Major comments:

1. Fig. 2, 3: The Western blotting data presented has serious flaws. The authors decided to separate the “control/ capsaicin-treatment” and “control/zinc-treatment” groups from the same gel. If lanes from a single gel image have been rearranged, image splicing should be clearly denoted by vertical black lines on the figure. The Figure legends and Methods should provide details of how the discontinuous images were made. A figure should not include composite images originating from different blots. I am really concerned about using the same data multiple times.

2. The authors persist that the major finding is “two bioactive nutrients activate the calcium signalling pathway and its downstream signalling molecules including CAMKK2, CREB, TORC1, regulate expression of target genes, and increase calcium-dependent glucose oxidation in C2C12 skeletal muscle cells”. However, they did not investigate whether CAMKK2, CREB, TORC1 signaling induce the glucose oxidation in C2C12 cells. The authors should examine the pharmacological experiments (inhibitors and/or agonists of CAMKK2, CREB, TORC1). Is each kinase inhibitor block the increase of capsaicin/Zn-induced Ca2+-dependent glucose oxidation in C2C12 skeletal muscle cells?

3. The authors answered that they used 20 µM ZnSO4 + 10 µM pyrithione sodium salt (Footnote 1). Does pyrithione sodium salt itself activate the signal pathways including calcium channels, TRPV1, CAMKK2, CREB, TORC1? Does pyrithione sodium salt affect glucose oxidation and intracellular cAMP levels in C2C12 cells? The authors should provide appropriate controls (0 µM ZnSO4 + 10 µM pyrithione sodium salt).

4. The authors did not answer the question why zinc increase calcium levels in a few minutes. They should explain the molecular mechanisms of Zn-induced calcium influx/release. In Taylor et al (2008), there is no data about Zn-induced calcium up-regulation. Villalobo et al. is just a review book. There is no evidence that zinc induces rapid increase in the cytosolic Ca2+ concentrations via EGF signaling. Since EGFR pathway activates transcriptional factors, isn't it impossible to increase the Ca2+ concentrations in a few minutes? Is zinc activate EGFR signaling in the absence of EGF? Is Zn-activated EGFR signaling directly activate calcium channels, ryanodine receptors or IP3 receptors within a few minutes? The authors should carefully discuss about these mechanisms.

Minor comments:

1. In Vahidi Ferdowi et al., Cells, 10, 1560, 2021, the author's group previously demonstrated that capsaicin activated CAMKK2-AMPK signaling pathway leading to increased glucose oxidation and ATP generation. In Figure 10 and Figure 11, the authors indicate the effect of capsaicin and zinc, independently. They do not use the inhibitors of TRPV1 or zinc transporter. The data presented are only partially convincing and too superficial to provide an appropriate understanding of the molecular mechanism of calcium signaling and the downstream events as the common pathway between capsaicin and zinc in glucose oxidation. The discussion and conclusion should be toned down to reflect the data. I think citing the studies performed by other researchers would support the conclusion of this manuscript.

2. Fig. 7 and 10, Line 288-296:  It should be kept in mind that BAPTA-AM inhibits many pathways and may exert other effect. It is unclear whether capsaicin/zinc is directly associated with calcium/CAMKK2/CREB/TORC1 signaling and glucose oxidation in this manuscript. The authors did not examine the pharmacological experiment but they added the manuscript (lines: 288-296). Here, the authors should cite the proper references and discuss the molecular mechanisms.  

3. Line 459: The authors should include the composition of Zn solution in the main text, not in the footnote. The composition of Zn solution is an important information for other researchers to investigate the reproducibility.

Reviewer 2 Report

The authors addressed my comments sufficiently